# Geometric Fragility in Alzheimer's Disease: Probing the Loss of Hippocampal Hierarchical Abstraction via Contrastive Point Cloud Modeling

**Yiquan Wang, Yuhan Chang, Jialin Zhang, Minnuo Cai, Jiayao Yan & Hongtian Zhao**
Xinjiang University
`ethan@stu.xju.edu.cn`   `zhaohongtian@xju.edu.cn`

**for the Alzheimer's Disease Neuroimaging Initiative**

## Abstract

While human spatial cognition relies on hierarchical abstraction to integrate local features, Alzheimer's disease (AD) pathology is often reduced to local volumetric atrophy. We propose geometric fragility as a distinct failure of global topological integrity within the hippocampus rather than a mere accumulation of local errors. Using a texture-invariant point cloud framework, we contrasted local feature aggregation with global hierarchical attention as computational probes for structural perception. The global paradigm demonstrated enhanced diagnostic sensitivity and exhibited cognitive resilience under simulated neuronal sparsity by maintaining structural recognition where local models suffered pattern collapse. Global modeling effectively unfolded the pathological manifold, revealing a linear trajectory of geometric degradation consistently correlated with clinical cognitive decline. These findings indicate that AD progression may be characterized as a systemic topological dissolution and suggest that effective biomarkers should prioritize hierarchical shape abstraction to accurately map the continuous transition from healthy cognition to dementia.

## 1 Introduction

Human spatial cognition relies fundamentally on the hierarchical abstraction of global topological skeletons rather than merely processing local textural details (Navon, 1977). Integrating local features into holistic structural descriptions is critical for the hippocampus to construct cognitive maps supporting spatial and episodic memory (O'keefe & Nadel, 1978). In the pathological evolution of Alzheimer's disease (AD), hippocampal degeneration is often quantified primarily through volumetric atrophy or a decrease in gray matter density (Villain et al., 2008; Xiao et al., 2023). Computationally, however, neurodegeneration may represent a fundamental geometric collapse—a loss of capacity to maintain topological integrity. If this hypothesis holds, a computational model capable of effectively diagnosing AD should possess hierarchical abstraction capabilities similar to the human visual system, enabling it to capture long-range geometric dependencies rather than merely aggregating local surface features.

Existing neuroimaging analyses predominantly rely on voxel-based morphometry or convolutional neural networks (Ashburner & Friston, 2000; Huang et al., 2023; Krüger et al., 2024). While these methods excel at feature extraction, their computational nature is typically constrained by local receptive fields, tending to make decisions by aggregating high-frequency textures or local shape variations (Geirhos et al., 2018; Ranabhat et al., 2025). This local-first processing paradigm reveals limitations when handling the hippocampus, an anatomical body with a highly non-rigid and complex manifold structure. This is particularly evident in the early pathological stages where the distortion of global structure has not yet manifested as significant local volume loss, making local models prone to missing these subtle topological signals. Furthermore, when facing data distribution

differences across centers or degradation in imaging quality, models that overly rely on local textural features tend to exhibit fragility, failing to maintain an understanding of the overall shape by filling in missing links in the way the human cognitive system does under sparse or noisy conditions.

To verify the geometric fragility hypothesis in Alzheimer's disease pathology, this study constructs a pure geometric point cloud framework that strips away texture information and retains only spatial coordinates. We utilize this framework as a computational probe to compare two distinct cognitive processing paradigms. The first is a graph convolutional network (DGCNN (Wang et al., 2019)) based on local neighborhood aggregation, simulating low-level local feature processing. The second is a hierarchical Point Cloud Transformer (LitePT (Yue et al., 2025)) based on self-attention mechanisms, simulating global integration processing capable of capturing long-range dependencies. As illustrated in Figure 1, these paradigms diverge significantly under computational stress such that when structural density is reduced to simulate atrophy, the local model suffers from neighborhood fracturing and processes the object merely as discrete fragments. In contrast, the global model demonstrates cognitive resilience by maintaining the topological persistence of the object, akin to identifying the identity of a chair through a sparse sketch (Biederman, 1987). We hypothesize that if the essence of AD is the collapse of global topological structure, then the global model should not only outperform the local model in diagnostic accuracy but should also demonstrate cognitive resilience similar to biological neural systems when facing data sparsification pressure. Our em-

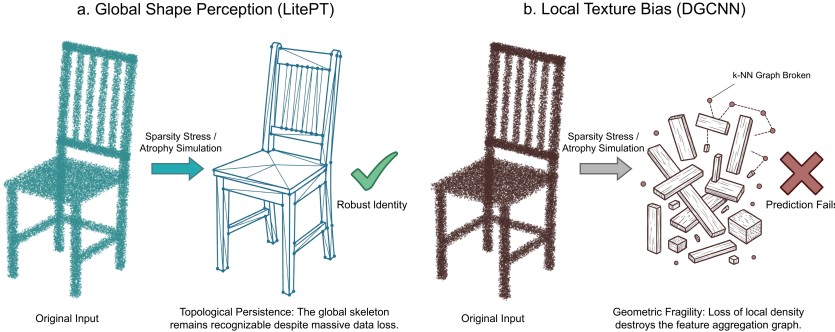

Figure 1: Geometric Fragility under Structural Degradation. (a) Global Shape Perception (LitePT) preserves topological integrity through hierarchical abstraction despite severe sparsification. (b) Local Texture Bias (DGCNN) exhibits representational collapse as neighborhood fracturing destroys the aggregation graph, illustrating the geometric fragility of local processing.

pirical findings validate these hypotheses by revealing a fundamental dissociation in how the two paradigms construct disease representations. Beyond demonstrating superior diagnostic precision at full resolution, the global architecture exhibited a form of structural persistence distinct from local aggregation methods. When subjected to simulated neuronal loss via sparsification, the local model suffered immediate representational collapse; in contrast, the global model maintained pathological sensitivity by anchoring onto the residual topological skeleton, effectively preserving structural recognition even when local cues were severely degraded. This behavioral resilience is rooted in the geometry of the learned feature space, where manifold analysis demonstrates that global abstraction successfully unfolds the disease trajectory. Unlike the entangled representation produced by local processing, this topological disentanglement linearizes the progression from health to dementia, situating Mild Cognitive Impairment as a coherent transitional bridge rather than a source of classification ambiguity. These results imply that AD pathology may be characterized more accurately as a dissolution of global shape integrity than as local texture variation, suggesting that computational probes prioritizing hierarchical abstraction are essential for mapping the continuous biological transition of neurodegeneration.

## 2 COMPUTATIONAL FRAMEWORK

To test whether AD pathology manifests as global topological dissolution rather than local feature accretion, we translated competing cognitive theories into distinct computational architectures within a unified pipeline (Figure 2). By designing two distinct Computational Observers constrained

by specific processing biases, we isolated geometric structural signals from confounding variables including tissue density or intensity, thereby creating a controlled environment to probe the specific mode of hippocampal failure.

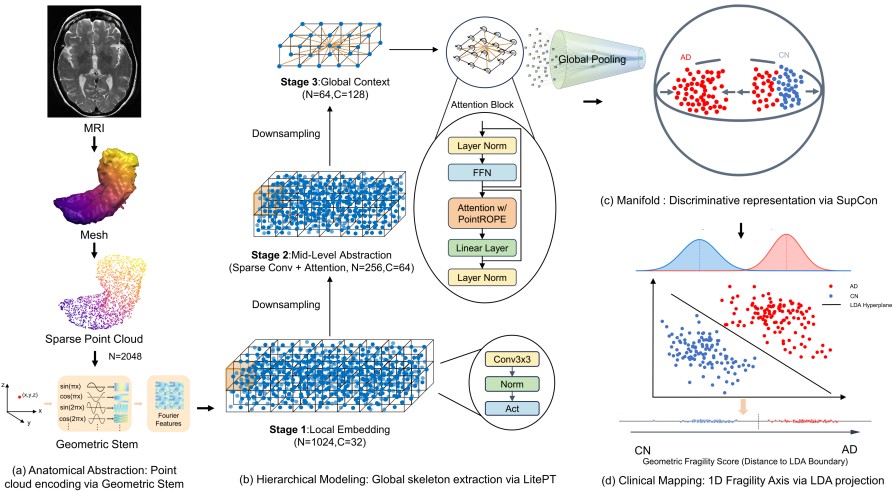

Figure 2: The Computational Pipeline for Geometric Fragility Probing. (a) Anatomical Abstraction: Hippocampi are converted into point clouds and encoded via Fourier features (Geometric Stem). (b) Hierarchical Modeling: The LitePT encoder captures global topological skeletons through self-attention mechanisms. (c) Manifold Learning: Supervised Contrastive Learning (SupCon) constructs a discriminative feature space. (d) Clinical Mapping: Linear Discriminant Analysis (LDA) projects the high-dimensional geometry onto a 1D Fragility Axis, serving as a diagnostic biomarker correlated with cognitive decline.

## 2.1 GEOMETRIC REPRESENTATION OF HIPPOCAMPAL ANATOMY

Traditional neuroimaging morphometry often confounds tissue density or gray scale values with geometric shape (Chung et al., 2003; Aird-Rossiter et al., 2026). To decouple these factors and force the computational model to learn solely from structural geometry, we converted volumetric MRI data into sparse point cloud representations. We transformed the hippocampus of each subject into a set of 2,048 spatial coordinate points $(x, y, z)$ normalized to a unit sphere. This excludes texture and density to isolate the underlying topological skeleton. By restricting input to spatial coordinates, we ensure diagnostic performance is attributed solely to geometric processing of the anatomical manifold. This representation aligns with cognitive theories suggesting that the brain encodes objects primarily through 3D spatial coordinates and surface topology rather than surface textural details (Marr, 2010).

## 2.2 MODELING LOCAL VERSUS HIERARCHICAL STRUCTURAL PROCESSING

We operationalized competing cognitive theories of shape perception by implementing two distinct neural architectures, each representing a unique mode of visual processing.

To simulate the local processing paradigm, we employed a Dynamic Graph Convolutional Network (DGCNN (Wang et al., 2019)). This architecture constructs representations by aggregating information exclusively from the $k$-nearest neighbors in feature space. Similar to early visual cortex processing or texture perception mechanisms, this model excels at detecting high-frequency surface variations such as local roughness or curvature but lacks an intrinsic mechanism to integrate these local features into a coherent whole across long spatial distances. It represents the hypothesis that AD diagnosis relies primarily on detecting local surface deformations.

Conversely, to simulate global hierarchical abstraction, we utilized the LitePT (Yue et al., 2025) architecture, a lightweight Point Cloud Transformer. Unlike graph-based methods, this model employs self-attention mechanisms that allow each point in the anatomical structure to attend to all others,

independent of spatial distance. This global receptive field enables the model to capture long-range topological dependencies and construct holistic structural descriptions, mirroring the part-whole integration process inherent in high-level visual cognition and hippocampal function. By embedding spatial coordinates with Fourier features (Tancik et al., 2020) prior to attention processing, the model prioritizes global context while preserving sensitivity to fine details. This architecture serves as a computational proxy to validate the hypothesis that the structural integrity of the hippocampal backbone is a primary biomarker of cognitive health.

## 2.3 Manifold Learning via Contrastive Optimization

To map these geometric representations into a clinically meaningful cognitive space, we adopted Supervised Contrastive Learning (Khosla et al., 2020). This learning objective is based on the principle of maximizing the representational difference between diagnostic groups while clustering structurally similar anatomies. By projecting high-dimensional geometric features onto a latent hypersphere, this method encourages the model to learn a shape manifold where the transition from normal cognition to dementia is encoded as a continuous geometric displacement (Bellmund et al., 2018). This manifold learning strategy is crucial to our analysis as it allows us to visualize the evolution of the shape of the disease and quantify the separability of diagnostic categories based on pure topological fingerprints.

## 2.4 Unfolding the Disease Trajectory via Linear Projection

While the contrastive learning phase constructs a high-dimensional geometric space where healthy and pathological structures are separable, quantifying the degree of degeneration requires a continuous metric rather than binary labels. To operationalize the concept of a disease continuum, we employed Linear Discriminant Analysis (LDA (Fisher, 1936)) not merely as a classifier but as a tool for Manifold Unfolding. By projecting the frozen high-dimensional representations learned by the model onto a low-dimensional axis that maximizes class separability, specifically the vector connecting the centroids of the healthy control group and the Alzheimer's disease group, we effectively linearized the complex pathological trajectory. This projection defines a Geometric Collapse Axis, enabling us to derive a scalar severity index for each subject based solely on hippocampal shape. This operationalization is critical for our subsequent analysis as it allows us to correlate discrete geometric positions with continuous cognitive scores, such as the MMSE, thereby verifying whether the model's internal representation of shape mirrors the behavioral decline observed in patients.

## 3 Results

### 3.1 Global Topology Enhances Diagnostic Accuracy

Under full-resolution (2,048 points) anatomical structures, the behavioral divergence between global and local processing paradigms suggests that the global model (LitePT) captures the categorical structure of the disease spectrum more effectively. While the difference in overall accuracy is notable (LitePT: 89.47% vs. DGCNN: 64.47%), a key observation lies in how the models construct boundaries between clinical stages (Figure 3a, c). As indicated by the fine-grained subtype analysis, LitePT exhibits behavior consistent with Categorical Perception (Goldstone & Hendrickson, 2010). It effectively maps the disease trajectory into distinguishable topological regions: healthy controls are largely anchored in the healthy subspace (92.9% predicted as CN); the intermediate spectrum from subjective memory complaints (SMC) to late MCI (LMCI) is consistently classified within the MCI category (84%–100% predicted as MCI); and a shift to the AD category is primarily observed when pathology reaches the severe stage (75.0% predicted as AD). Notably, the model shows clear separation between the two extremes of the spectrum.

In comparison, the local model (DGCNN) appears to show reduced discriminative capacity, tending to project the clinical continuum toward a central state (Figure 3b, d). This phenomenon, which we term Representational Stagnation, is characterized by predictions favoring the MCI category across stages. The confusion matrix reveals a pronounced majority class bias, where 92.9% of healthy controls and 68.8% of Alzheimer's patients are misclassified as MCI. This suggests that without global topological context, the local geometric features of healthy and diseased hippocampi may

appear similar to the intermediate state. Consequently, the local operator appears to struggle with establishing effective decision boundaries, treating healthy, transitional, and dementia samples as structurally similar states.

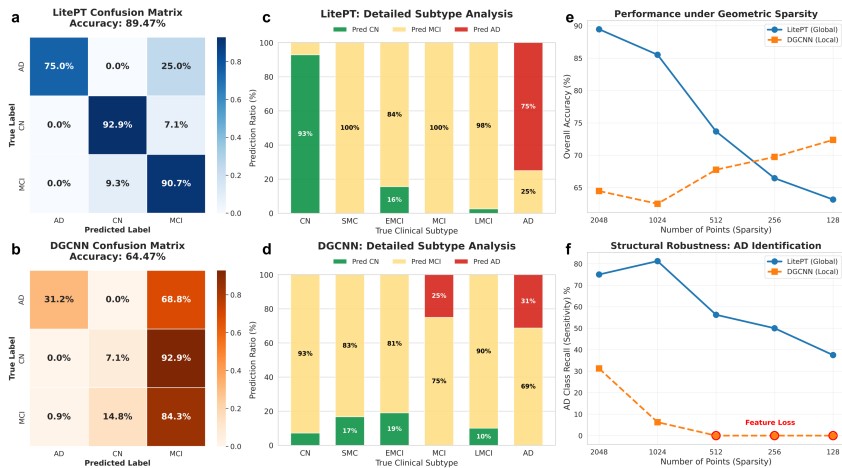

Figure 3: Comparative Diagnostic Performance of LitePT (Global) vs. DGCNN (Local). (a-b) Confusion Matrices at full resolution. (c-d) Predictions mapped to clinical subtypes; LitePT (c) shows clear phase transitions, while DGCNN (d) exhibits stagnation in the MCI spectrum. (e-f) Geometric Sparsification Stress Test comparing Overall Accuracy (e) and AD Recall (f) as point density decreases.

## 3.2 PROBING GEOMETRIC RESILIENCE: THE ILLUSION OF ACCURACY UNDER SPARSITY

To deeply validate whether the model has truly learned the topological essence of hippocampal degeneration rather than merely fitting local statistical regularities, we designed a stress test simulating cognitive load. By progressively reducing the density of the input point cloud, we deprived the model of available local spatial information to examine the performance of the two computational paradigms under sparse information conditions. This process is analogous to testing the Gestalt completion capability of biological visual systems when facing degraded inputs, specifically whether they can reconstruct the overall geometric skeleton in the absence of local cues.

Table 1 reveals a cognitive paradox: when spatial resolution was compressed (2,048 to 128 points), DGCNN exhibited a counter-intuitive accuracy increase (64.47% to 72.37%). Inspection of class metrics, however, identifies this as an artifact of mode collapse rather than genuine robustness. Under extreme sparsity, the fracturing of local neighborhoods caused the DGCNN to lose sensitivity to pathological features entirely resulting in an Alzheimer's Disease Recall of 0.00%. The superficial rise in accuracy was driven largely by a pathological bias towards the majority class evidenced by the model assigning nearly all samples to MCI while failing to construct a discriminative feature space for the extremes of the spectrum. Conversely, the LitePT model based on global attention mechanisms demonstrated graceful degradation characteristics similar to biological intelligence. Even with information scarcity of only 128 points, LitePT maintained an AD recall of 37.50% and successfully identified a significant portion of severe patients. This indicates the model has internalized the global topological schema rather than mechanically relying on dense local coordinate accumulation. Utilizing the long-range dependency capture ability of the self-attention mechanism, the model establishes geometric associations across space between discrete sparse points. Just as a human observer can identify object contours from a sketch of a few strokes, the global model effectively restores the overall morphological description through the residual topological skeleton. This capacity to maintain structural constancy in low signal-to-noise environments suggests that global hierarchical abstraction is a critical mechanism for robust diagnosis.

Table 1: Diagnostic performance metrics under progressive structural sparsification.

| Model & Sparsity | Overall Acc. | Macro F1 | AD Recall | AD Precision | CN Recall | CN Precision | MCI Recall | MCI Precision |
|---|---|---|---|---|---|---|---|---|
| LitePT (2048) | 89.47% | 0.865 | 75.00% | 100.00% | 92.86% | 72.22% | 90.74% | 94.23% |
| LitePT (1024) | 85.53% | 0.819 | 81.25% | 86.67% | 82.14% | 63.89% | 87.04% | 93.07% |
| LitePT (512) | 73.68% | 0.650 | 56.25% | 90.00% | 46.43% | 40.62% | 83.33% | 81.82% |
| LitePT (256) | 66.45% | 0.533 | 50.00% | 57.14% | 28.57% | 27.59% | 78.70% | 77.98% |
| LitePT (128) | 63.16% | 0.456 | 37.50% | 46.15% | 17.86% | 18.52% | 78.70% | 75.89% |
| DGCNN (2048) | 64.47% | 0.438 | 31.25% | 83.33% | 7.14% | 11.11% | 84.26% | 71.09% |
| DGCNN (1024) | 62.50% | 0.344 | 6.25% | 33.33% | 14.29% | 18.18% | 83.33% | 70.87% |
| DGCNN (512) | 67.76% | 0.289 | 0.00% | 0.00% | 3.57% | 14.29% | 94.44% | 70.83% |
| DGCNN (256) | 69.74% | 0.295 | 0.00% | 0.00% | 3.57% | 25.00% | 97.22% | 71.43% |
| DGCNN (128) | 72.37% | 0.388 | 0.00% | 0.00% | 21.43% | 75.00% | 96.30% | 73.24% |

## 3.3 Unfolding the Disease Manifold

The geometric schemas generated by the two processing paradigms reveal fundamental differences in how neurodegenerative patterns are encoded within the latent feature space. The DGCNN model, constrained to local feature aggregation, constructs a geometrically entangled manifold that fails to resolve the ambiguity of early pathology. As illustrated in the left panel of Figure 4a, the representation appears entangled and disordered where early-stage MCI samples are rendered topologically indistinguishable from healthy controls. While the model captures a rudimentary continuity, this local connectedness lacks the necessary feature decoupling for clinical stratification. This representational failure is quantitatively confirmed by the LDA density projection in the left panel of Figure 4b, where the separation distance between the centroids of the healthy and Alzheimer's groups is merely 2.27. The resulting decision boundary exhibits a chaotic distribution, indicating that reliance on local geometric textures alone is insufficient to disentangle pathological degeneration from normal anatomical variance.

In distinct contrast, the LitePT model leverages global hierarchical abstraction to achieve a significant Manifold Unfolding effect. The right panel of Figure 4a demonstrates that the model successfully straightens the pathological trajectory into a linear evolution axis, revealing a clear directional divergence from the cluster of healthy controls toward the severe disease state. This geometric disentanglement is substantiated by the LDA analysis in the right panel of Figure 4b, which yields a markedly increased inter-class separation distance of 7.94 compared to the local model. Rather than forcing an artificial binary separation, LitePT preserves the biological coherence of the disease spectrum by accurately situating the MCI group within the transitional space between health and dementia. These findings suggest that global topological unfolding is the requisite computational mechanism for mapping the continuous degradation of hippocampal shape onto distinct, yet biologically connected, clinical categories.

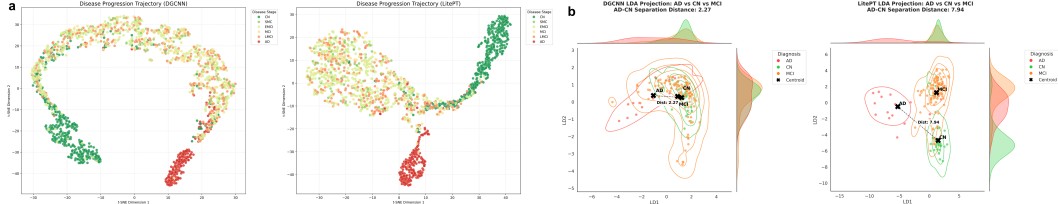

Figure 4: Visualization of the learned feature space. (a) t-SNE manifold visualizations comparing the entangled representation of the local DGCNN model (left) against the unfolded trajectory of the global LitePT model (right). (b) Linear Discriminant Analysis (LDA) density projections quantifying separability, showing limited distinction in the local model (left) versus robust disentanglement in the global model (right).

## 3.4 Alignment between Geometric Representations and Cognitive Behavior

To evaluate whether the learned geometric features possess biological relevance beyond merely fitting training labels, we examined the relationship between the feature space and external clini-

cal metrics. We performed a regression analysis using the model-derived projection axis against Mini-Mental State Examination scores (Goldstone & Hendrickson, 2010). Since this clinical gold standard was withheld during training the resulting correlation serves as an objective test of whether the model's latent space reflects actual cognitive status.

The comparison reveals a notable divergence in the manner by which the two architectures map anatomical shape to function. The LitePT model based on global abstraction exhibited a moderate negative correlation with clinical scores ($r = -0.544$) whereas the local DGCNN model showed a weaker association ($r = -0.286$). More importantly the global model appeared to better structure the intermediate Mild Cognitive Impairment group as a transitional state distinct from the noise observed in the local model's projections. While the trajectory is not perfectly linear the global approach provided a more coherent ordering of subjects that roughly tracks the continuous spectrum of neurodegeneration.

This difference implies that feature extraction limited to local neighborhoods may struggle to decouple normal anatomical variance from pathology-driven decline. The scatter observed in the local model suggests that without global topological constraints it is difficult to distinguish benign surface irregularities from relevant atrophy. These findings indicate that the overall hippocampal skeleton may offer a more reliable structural proxy for cognitive reserve than local surface details alone.

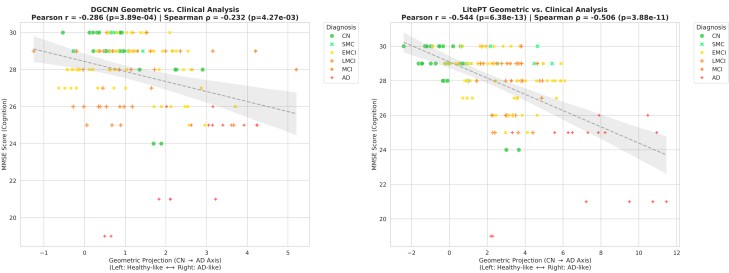

Figure 5: Regression analysis of model-derived geometric projections (x-axis) versus clinical MMSE scores (y-axis). The local DGCNN model (left) exhibits a weak correlation ($r = -0.286$) with chaotic class separation. In contrast, the global LitePT model (right) demonstrates a strong linear mapping ($r = -0.544$), accurately positioning MCI subgroups (yellow/orange) as a transitional bridge between Healthy Controls (green) and AD (red).

## 3.5 PERCEPTUAL CONSTANCY AND STRUCTURAL INVARIANCE ACROSS IMAGING DOMAINS (OASIS-1 ANALYSIS)

A core feature of the cognitive system is perceptual constancy, the ability to maintain stable representations of object structure despite drastic changes in sensory input conditions such as lighting, perspective, or noise levels (Walsh & Kulikowski, 1998; Garrigan & Kellman, 2008). In the context of neuroimaging computing, domain shifts caused by differences in scanning equipment hardware and acquisition parameters across different data centers constitute the ultimate test of a computational model's perceptual constancy. If a model has truly learned the pathological geometric essence of the hippocampus rather than merely fitting statistical noise in the training set, it should possess generalization capabilities across imaging protocols. To test this hypothesis, we applied the model constructed on the ADNI dataset (Jack Jr et al., 2008) directly to a completely independent external dataset, OASIS-1 (Marcus et al., 2007), for zero-shot cross-domain inference without any retraining.

The results reveal distinct behavioral patterns of the two cognitive paradigms when facing environmental uncertainty. As detailed in Table 2, the DGCNN model based on local feature aggregation suffered severe representational failure. Although it appeared to retain high recall on the Alzheimer's disease (AD) category (28.33%), this was in fact a pathological prediction bias. In-depth analysis revealed that the model's ability to identify the healthy control group (CN) was almost completely lost, with a recall rate of only 9.33%. This extreme imbalance led to a significant decrease in its Macro F1-Score to 0.284, revealing that the model did not establish effective decision boundaries but degenerated into a biased heuristic tending to judge unknown samples as pathological states. This suggests that the local model is highly dependent on specific voxel textures or scanner noise in

Table 2: Zero-shot cross-domain generalization performance on the external OASIS-1 dataset.

| Model | Overall Accuracy | Macro F1 | CN Recall | AD Recall | MCI Recall |
|---|---|---|---|---|---|
| LitePT | 41.45% | 0.379 | 30.97% | 23.33% | 69.29% |
| DGCNN | 30.77% | 0.284 | 9.33% | 28.33% | 72.86% |

the training data, and when these non-essential cues change in the new imaging domain, the model loses the baseline for distinguishing health from disease.

Conversely, the LitePT model, equipped with global hierarchical abstraction capabilities, demonstrated structural resilience similar to biological visual systems. Despite facing huge distribution differences, the model maintained an overall accuracy of 41.45%, and crucially, its Macro F1-Score reached 0.379, significantly outperforming the local model. The advantage in this metric stems from LitePT maintaining relatively balanced discriminative power across clinical categories, particularly retaining a recall rate of 30.97% in the healthy control group, proving its success in resisting cross-domain noise interference that misclassifies healthy samples as pathological. This indicates that LitePT, through self-attention mechanisms, successfully stripped away shallow texture information related to equipment and anchored onto the more stable global topological skeleton. This high-dimensional structural invariant of the overall hippocampal geometry proves to be a biomarker with stability across different imaging protocols (Gibson, 1979), strongly supporting global hierarchical abstraction as a key cognitive mechanism for achieving robust AD diagnosis.

## 4 DISCUSSION & CONCLUSION

The computational probe constructed in this study reveals the core pathological feature of Alzheimer's disease at the neuroanatomical level: hippocampal lesions are not essentially simple local tissue loss but a systemic collapse of the capacity for hierarchical abstraction of the 3D topological skeleton. The divergence in performance between models with different hierarchical abstraction capabilities challenges the traditional morphometric definition of pathological features. The systemic failure of models based on local neighborhood aggregation implies that in the early stages of disease progression, the local microscopic geometric texture of the hippocampus may not have undergone specific changes, or such local variations are masked by inherent inter-individual anatomical noise. The reason the model with global attention mechanisms can successfully establish clear diagnostic boundaries is that it simulates processing mechanisms similar to Gestalt Perception in biological visual systems (Fu et al., 2025), specifically by integrating non-adjacent spatial nodes to construct a holistic structural description independent of local details. This finding supports the topological-first pathological hypothesis, suggesting that hippocampal functional decline manifests first as the disintegration of global geometric constraints rather than the isolated destruction of local tissue.

This computational perspective on structural integrity further provides a theoretical basis for understanding the cognitive reserve of the nervous system. In sparsification stress tests simulating accelerated neuronal loss (Terry et al., 1991; Cai et al., 2025), the local model rapidly fell into mode collapse, while the global model demonstrated graceful degradation characteristics similar to biological neural systems. This difference reflects how the brain may maintain function in the context of physical damage where cognitive robustness relies on the internalization of high-dimensional topological skeletons rather than the retention of high-frequency details. The global model's ability to reconstruct the overall shape through pattern completion under conditions where only a few key points remain mirrors computationally the brain's compensatory mechanism of maintaining macroscopic network topological properties by reorganizing residual connections in the face of synaptic pruning or neuronal death. Therefore, maintaining long-range geometric associations is key to resisting the information entropy increase caused by atrophy.

Finally, the model's perceptual constancy across imaging domains validates global topological features as biological invariants. Human cognition maintains object recognition across changing environments; our global model's robustness against unknown imaging protocols computationally mirrors this constancy. Conversely, the local model's overfitting to specific distributions suggests that

without high-level abstraction, perceptual systems are easily deceived by environmental noise. This study confirms that only models possessing hierarchical abstraction can capture concealed pathological signals and demonstrate resilience akin to biological intelligence. The geometric manifold we constructed successfully links abstract anatomical features with clinical behaviors, demonstrating that hippocampal geometry serves as a functional mapping. These findings suggest future research should prioritize geometric orders capable of global binding, as their maintenance or collapse defines the integrity of cognitive health.

## MEANINGFULNESS STATEMENT

A meaningful representation of life captures biological resilience: the ability to maintain global integrity despite local decay. We propose that cognitive health relies on hierarchical geometric abstraction to perceive holistic shapes beyond fragmented details. Our work frames Alzheimer's disease as a collapse of this global topological coherence within the hippocampus. By demonstrating that global processing paradigms mirror human resilience under stress, we show that modeling life requires capturing long-range dependencies rather than merely aggregating local features.

## FUNDING

This work was supported by the Xinjiang Uygur Autonomous Region Natural Science Foundation Youth Project (No. 2025D01C276), the Xinjiang Tianchi Talent Program "Robust Perception and Restoration of Low-Quality Visual Content" (No. 51052501848), the National College Student Innovation Training Program (Project No. 202510755108), and the Xinjiang Uygur Autonomous Region College Student Innovation Training Program (Project No. S202510755168).

## ACKNOWLEDGMENTS

This work was supported in part by the Computing and Data Center of Xinjiang University. We acknowledge the computing resources and technical support provided by the Computing and Data Center of Xinjiang University.

Data used in preparation of this article were obtained from the Alzheimer's Disease Neuroimaging Initiative (ADNI) database (adni.loni.usc.edu). As such, the investigators within the ADNI contributed to the design and implementation of ADNI and/or provided data but did not participate in the analysis or writing of this report. A complete listing of ADNI investigators can be found at: http://adni.loni.usc.edu/wp-content/uploads/how_to_apply/ADNI_Acknowledgement_List.pdf

Data collection and sharing for the Alzheimer's Disease Neuroimaging Initiative (ADNI) is funded by the National Institute on Aging (National Institutes of Health Grant U19AG024904). The grantee organization is the Northern California Institute for Research and Education. In the past, ADNI has also received funding from the National Institute of Biomedical Imaging and Bioengineering, the Canadian Institutes of Health Research, and private sector contributions through the Foundation for the National Institutes of Health (FNIH) including generous contributions from the following: AbbVie, Alzheimer's Association; Alzheimer's Drug Discovery Foundation; Araclon Biotech; Bio-Clinica, Inc.; Biogen; Bristol-Myers Squibb Company; CereSpir, Inc.; Cogstate; Eisai Inc.; Elan Pharmaceuticals, Inc.; Eli Lilly and Company; EuroImmun; F. Hoffmann-La Roche Ltd and its affiliated company Genentech, Inc.; Fujirebio; GE Healthcare; IXICO Ltd.; Janssen Alzheimer Immunotherapy Research & Development, LLC.; Johnson & Johnson Pharmaceutical Research & Development LLC.; Lumosity; Lundbeck; Merck & Co., Inc.; Meso Scale Diagnostics, LLC.; NeuroRx Research; Neurotrack Technologies; Novartis Pharmaceuticals Corporation; Pfizer Inc.; Piramal Imaging; Servier; Takeda Pharmaceutical Company; and Transition Therapeutics.

## DATA AND CODE AVAILABILITY

The source code for this study is available on GitHub at https://github.com/wyqmath/AD_Hierarchical_Abstraction.

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
