# OpenReview forum: "Geometric Fragility in Alzheimer's Disease: Probing the Loss of Hippocampal Hierarchical Abstraction via Contrastive Point Cloud Modeling"
_ICLR.cc/2026/Workshop/LMRL — ICLR 2026 Workshop LMRL Poster_

### Official Review · Reviewer_74jg · 2026-02-17
**Good results, another pass over manuscript required**

**Rating:** 6
**Confidence:** 3

**Review:**

The work encodes the hippocampus spatial structure as point clouds encoded using Fourier features, in the process removing all information apart from the pure spatial coordinates. These purely spatial features are used to train models to classify different Alzheimers disease states. Models are trained using supervised contrastive learning to maximise difference between healthy and unhealthy while clustering anatomies that are structurally similar. Two models are compared, a dynamic graph convolution neural network DGCNN and a transformer. This enables comparison of local vs global feature aggregation. It is found that the transformer has higher diagnostic performance, robustness to sparsity and is more amenable to linear discriminant analysis, enabling a single vector that maps between healthy and disease states to be created.  It is further shown that this vector correlates somewhat with Mini-Mental State Examination scores and that the transformer model somewhat generalises to an independent dataset.

With the caveat that I am not a neuroscience/Alzheimers researcher I think that overall this work is interesting and valuable. However, I find the write up to be quite confusing and muddled at times. I find some of the references to human spatial cognition to be confusing, sometimes this seems to refer to the structure of the hypothalamus that the models use as input features, sometimes to the abstract method by which the brain understands images and sometimes to the inductive bias of the transformer model. Perhaps my naivety to this field is causing my confusion, but it feels like alot of the text was AI generated and then not fully rewritten to remove all instances of confusion, in either case I think clarity could be improved overall.

Further, some of the text in the results seems counter to what is shown in the figures (another thing I have found that LLMs, especially Claude specifically can do). e.g. lines 282-283 "As illustrated in the left panel of Figure 4a, the representation appears entangled and disordered where early-stage MCI samples are rendered topologically indistinguishable from healthy controls", when looking at 4a I see good separation between the CN and AD classes, comparable to 4b. I think there needs to be some more explanation of why the results in 4b are better than 4a as is claimed. I understand that 4b is more amenable to LDA, but why this is so important should be further explained i.e. why can't I just project the data down to the curved trajectory between CN and AD clearly shown in 4a. Also the text on figure 4 is tiny.

I think although the work here and the results are valuable and interesting, but another pass over the manuscript is required.


Pros:
- Good investigation that demonstrates local feature aggregation leads to better diagnostic outcomes for transformers

Cons:
- text is confusing at times

---

### Official Review · Reviewer_fXEM · 2026-02-25
**Conceptually Novel Approach to AD Progression with Methodological Concerns**

**Rating:** 3
**Confidence:** 3

**Review:**

### Summary

This work presents a novel method for analyzing Alzheimer's Disease (AD) progression based on changes in the global structure of the hippocampus (HPC). The authors state that existing methods for quantifying AD primarily focus on local structural atrophy, typically assessed using convolutional neural networks (CNNs). They argue that these conventional methods are constrained by local receptive fields, failing to utilize global features and making them susceptible to technical batch variations. Furthermore, they posit that local structural atrophy manifests in later stages of AD, whereas changes to the global HPC structure occur earlier.

Motivated by this, the authors hypothesize that AD might manifest as changes in the global structure of the HPC before volumetric grey matter atrophy occurs, proposing global structure as an early biomarker. To test whether AD manifests as an accumulation of local versus global changes, they evaluate two point cloud learning methods: DGCNN (representing the local paradigm) and LitePT (representing the global paradigm), concepts drawn from competing cognitive theories of shape perception. They find that the LitePT model is better able to classify intermediate stages of mild cognitive impairment (MCI) and *could* be used as a way to model disease progression.

### Quality

The methodological quality is mixed; while the use of an independent dataset strengthens generalizability, the validity of the results is undermined by contradictory claims regarding texture reliance, unsupported assertions drawn from low-dimensional embeddings (t-SNE), and the potentially unfair evaluation of local models on heavily downsampled inputs.

### Clarity

The manuscript suffers from critical gaps in reproducibility, specifically regarding the pipeline from volumetric MRI to point clouds, alongside the use of vague terminology ("significant," "cognitive resilience") and undefined abbreviations (e.g., "CN" is never explicitly defined) that obscure the core narrative.

### Originality

The work demonstrates strong conceptual originality by reframing Alzheimer's progression through the lens of cognitive shape perception theories, proposing global geometric structure as an early biomarker rather than relying on standard volumetric gray matter atrophy.

### Pros

- The novel use of point clouds effectively isolates structural information to directly compare global versus local paradigms without the confound of local texture.
- Evaluating the models on a separate dataset provides robust validation against technical batch effects like scanner noise and patient variation.

### Cons

- The introduction abruptly jumps from human spatial cognition to AD pathology, introducing concepts like "cognitive resilience" and the "geometric collapse" hypothesis without sufficient biological justification or supporting references.
- The procedure for converting volumetric MRI meshes to point clouds is insufficiently explained, leaving it unclear if downsampling inherently removes texture or if explicit constraints were required to achieve a texture-invariant representation.
- Essential model details are missing, including the selection criteria for DGCNN's $k$ parameter and an explanation of the mechanism by which Fourier features enable LitePT to prioritize global context.
- Evaluating the local-first DGCNN on heavily downsampled 2048-point clouds may unfairly penalize its performance by removing the high-resolution information it natively relies on.
- Figure 1 relies on circular reasoning to assert paradigm divergence rather than providing quantitative evidence of model performance on sparse structures.
- The authors contradict their own structural-only premise by claiming the local model's poor performance on OASIS-1 is due to specific voxel textures, despite stating that the preprocessing abstracts such local details.
- Claims regarding disease progression trajectories and topological indistinguishability rely heavily on qualitative t-SNE embeddings rather than rigorous quantitative measures like the provided LDA projections.
- The manuscript uses overly assertive language for an exploratory study and frequently employs ambiguous modifiers like "significant" without clarifying if this refers to statistical significance.
- The MMSE is used to connect results to clinical outcomes without a brief description outlining what the score quantifies, whether it relies on self-reporting, and what potential confounders might affect its reliability.
- The abbreviation "CN" is used for the healthy control group before being defined, creating ambiguity that should be resolved by standardizing terminology with cited literature where CN explicitly means "cognitively normal."

---

### Official Review · Reviewer_caPS · 2026-02-25
**Global vs Local Representations in AD**

**Rating:** 6
**Confidence:** 3

**Review:**

## Sumary
This paper proposes the hypothesis that Alzheimer’s disease (AD) reflects a collapse of global topological integrity in hippocampal geometry rather than local atrophy. To test this, the authors convert MRI to point clouds and train two models on this data: DGCNN, representing local neighborhood aggregation, and LitePT, representing global hierarchical abstraction. Models are trained using Supervised Contrastive Learning. A 1D “Geometric Collapse Axis” is then derived via LDA to model disease progression. The authors evaluate classification accuracy, robustness under progressive sparsification, correlation with MMSE scores and zero-shot cross-domain transfer (ADNI → OASIS-1). They conclude that global hierarchical modeling better captures disease structure and exhibits greater resilience.

## Strengths
- Clear and conceptually interesting framing (global vs local).
- Clean data processing using coordinate-only point clouds.
- The sparsification stress test.
- Inclusion of cross-domain evaluation.
- Analysis of the embedding space with LDA for interpretability.

## Weaknesses
- The claim that AD represents a global collapse is too strong. Currently this is compared with two models of different architectures, so it is not possible to distinguish the effect of the model choice and model size from the true biological signal.
- A lot of methodological details are missing, e.g. model sizes, dataset sizes, class distribution, train/test split, how the split was done, training details.
- No statistical testing, no confidence intervals reported for accuracy (e.g. from cross-validation).
- Low performance on OASIS-1 data limits the applicability of the models in real clinical settings.

## Questions
- Is it possible to restrict the transformer model to only local neighborhood to be able to draw conclusions not confounded by the model selection?

---

### Meta-Review · Area_Chair_kuZc · 2026-02-28

**Recommendation:** Accept (Poster)
**Confidence:** 4

**Metareview:**

The reviewers raise a number of concerns that should be addressed by the authors, but the majority felt it met the acceptance criterion so I think it is work discussing at the workshop.

---

### Decision · Program_Chairs · 2026-03-02

**Decision:**

Accept (Poster)

**Comment:**

Please see the meta-review.